| Open Peer Review | Computational Biology | Matters Arising

# Compositional transformations can reasonably introduce phenotype-associated values into sparse features

George I. Austin,[1,2] Tal Korem[2,3]

**ABSTRACT** Gihawi et al. (mBio 14:e01607-23, 2023, https://doi.org/10.1128/mbio.01607-23) argued that the analysis of tumor-associated microbiome data by Poore et al. (Nature 579:567-574, 2020, https://doi.org/10.1038/s41586-020-2095-1) is invalid because features that were originally very sparse (genera with mostly zero read counts) became associated with the phenotype following batch correction. Here, we examine whether such an observation should necessarily indicate issues with processing or machine learning pipelines. We show counterexamples using the centered log ratio (CLR) transformation, which is often used for analysis of compositional microbiome data. The CLR transformation has similarities to voom-SNM, the batch-correction method brought into question by Gihawi et al., and yet is a sample-wise operation that cannot, in itself, "leak" information or invalidate downstream analyses. We show that because the CLR transformation divides each value by the geometric mean of its sample, common imputation strategies for missing or zero values result in transformed features that are associated with the geometric mean. Through analyses of both synthetic and vaginal microbiome data sets, we demonstrate that when the geometric mean is associated with a phenotype, sparse and CLR-transformed features will also become associated with it. We re-analyze features highlighted by Gihawi et al. and demonstrate that the phenomenon of sparse features becoming phenotype-associated can also be observed after a CLR transformation, which serves as a counterexample to the claim that such an observation necessarily means information leakage. While we do not intend to address other concerns regarding tumor microbiome analyses, validate Poore et al.'s results, or evaluate batch-correction pipelines, we conclude that because phenotype-associated features that were initially sparse can be created by a sample-wise transformation that cannot artifactually inflate machine learning performance, their detection is not independently sufficient to demonstrate information leakage in machine learning pipelines. Microbiome data are multivariate, and as such, a value of 0 carries a different meaning for each sample. Many transformations, including CLR and other batch-correction methods, are likewise multivariate, and, as these issues demonstrate, each individual feature should be interpreted with caution.

**IMPORTANCE** Gihawi et al. claim that finding that a transformation turned highly sparse (mostly zero) features into features that are associated with a phenotype is sufficient to conclude that there is information leakage and to invalidate an analysis. This claim has critical implications for both the debate regarding The Cancer Genome Atlas (TCGA) cancer microbiome analysis and for interpretation and evaluation of analyses in the microbiome field at large. We show by counterexamples and by reanalysis that such transformations can be valid.

**KEYWORDS** microbiome, compositional data analysis, machine learning, imputation

**Peer Reviewer** Justin D. Silverman, The Pennsylvania State University, University Park, Pennsylvania, USA

Address correspondence to Tal Korem, tal.korem@columbia.edu.

The authors declare no conflict of interest.

See the funding table on p. 10.

See the companion article at https://doi.org/10.1128/mBio.01607-23.

In two critiques published last year (1, 2), Gihawi et al. raised several concerns regarding an analysis of the tumor microbiome in The Cancer Genome Atlas (TCGA) data (3). Among these concerns, they highlight several taxa that have mostly zero counts in raw data but that following batch correction have values that are correlated with specific tumor types. They claim that this is sufficient to indicate information leakage (1, 2). This implies a general principle that we wished to tackle: is finding that a transformation turns a sparse feature into a feature that is associated with a phenotype sufficient to conclude that it introduced information leakage (4)?

We demonstrate that this is not the case by constructing a counterexample, which involves no information leakage but still produces the same observation. To this end, we use the centered log ratio (CLR) transformation (5), a transformation that is simple, widely used in the microbiome field, bears similarities to the transformation in question (6, 7), and, most importantly, is a sample-wise transformation that does not use any metadata labels and cannot, in itself, induce information leakage.

Most microbiome data sets have an arbitrary total read count that is not reflective of sample properties and should therefore be interpreted as relative abundances that are compositional (5, 8, 9). Compositional data violate many of the assumptions underlying common data analysis strategies and, for example, exhibit a negative correlation bias and sub-compositional incoherence (5, 9). As components of compositional data can only be understood relative to one another, compositional transformations will generally transform them with respect to a reference—in the case of CLR, with respect to the geometric mean of the sample (10). Additionally, as most compositional transformations involve the use of a logarithm, they cannot handle zeros. Therefore, a common strategy is to introduce pseudocounts prior to the transformation. Previous studies have highlighted the difficulty in interpreting compositionally transformed features, which tend to contain information determined by the rest of the sample (9–11). As a result, some have advocated for alternatives to normalization-based methods, such as quantifying scale uncertainties (12, 13), although CLR and other normalization methods remain commonplace. In general, these intricacies have complicated inference from microbiome data, which recent work has demonstrated to be an ongoing challenge (14).

Here, we start by showing via simulations that as CLR-transformed sparse features are, by definition, negatively correlated with the geometric means of their corresponding samples, they could be associated with a phenotype that is itself associated with the geometric mean. Furthermore, as the geometric mean is related to α diversity (15), we demonstrate, through an analysis of a vaginal microbiome study (16), that CLR-transformed sparse features can be associated with a phenotype in cases where the sample α diversity is also associated with it. Finally, we reanalyze examples of sparse features highlighted by Gihawi et al. (1) and show that in cases where the α diversity was also associated with the phenotype in question, the phenomenon observed by Gihawi et al. could also be observed when performing CLR transformation. Thus, we show that the CLR transformation, which does not utilize any phenotype labels or information from other samples and therefore has no risk of independently inducing information leakage, can reasonably transform sparse features to non-sparse features with phenotype associations. While we recommend caution with univariate analyses or other attempts to interpret these transformed sparse features or to assign them any biological significance, their existence does not, on its own, discredit a data processing pipeline or downstream machine learning models as they are the product of a multivariate transformation.

## CLR-transformed sparse features are strongly associated with a phenotype in a simulated data set

We simulated a data set with 100 samples, 50 with a positive label and 50 with a negative label (Fig. 1a and b). By construction, the data had one completely empty feature and higher geometric means for the positive samples (see "Methods") with perfect separation from the negative samples (median [range] geometric means of 0.018 [0.012–0.022]

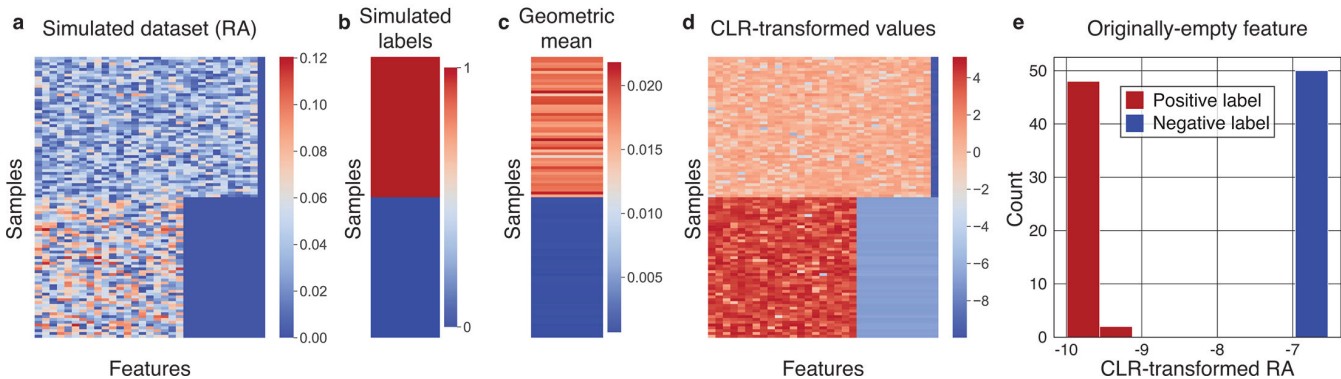

**CLR transformation of an empty feature can reasonably induce an association with phenotype**

**FIG 1** CLR transformation following a pseudocount can make a sparse feature highly predictive. (a, b) Heatmaps visualizing the relative abundances (RA) (a) and labels (b) of our simulated data set, in which there are 10 features present only in the "positive" samples and one "empty" feature that is zero for all samples (shown in the rightmost column). (c) The geometric mean of each sample, which is by construction highly correlated with the labels. (d) Heatmap of the relative abundances (a) after a CLR transformation in which they are divided by the geometric mean (c) following the addition of a pseudocount. (e) Histogram of the CLR-transformed empty feature (rightmost column of d) which was originally all zero in the simulated data set (rightmost column of panel a). The data set was constructed to have different geometric means between the samples with synthetic positive and negative labels. Thus, a CLR transform creates a perfect separation between this feature's values in the positive and negative samples (Mann–Whitney $U$ $P < 10^{-12}$).

and 0.00090 [0.00069–0.00099] for positive and negative samples, respectively; Mann–Whitney $U$ $P < 10^{-12}$; Fig. 1c). As commonly done prior to CLR transformation, we added a pseudocount of $10^{-6}$ to all values (see "Methods"). For the empty feature, dividing the pseudocounts by the geometric mean of each sample during CLR transformation created large differences between the positive and negative samples (median [range] values −9.80 [−9.99 to −9.41] vs −6.80 [−6.90 to −6.53], respectively; $P < 10^{-12}$; Fig. 1d and e). Of note, the same results hold when the added feature is non-zero ($P = 2.1 \times 10^{-17}$; Fig. S1a and b), partially zero ($P = 1.3 \times 10^{-5}$; Fig. S1c and d), as well as for a range of pseudocount values ($P < 0.01$ for pseudocounts ranging from $10^{-8}$ to 0.01; Fig. S2a). Based on this perfect separation, in most reasonable machine learning models, this feature would be highly predictive of our synthetically constructed label. However, this does not discredit the validity of either the models or of the transformation we performed. The CLR transformation is a sample-wise transformation, which does not observe labels or values from other samples. It therefore cannot "leak" information from the labels or a test set.

## CLR-transformed sparse features can be associated with clinical phenotypes

After observing that the CLR transformation of empty features can be informative for predicting an *in silico* phenotype label, we sought to demonstrate that similar conclusions can hold in real microbiome data sets. To do this, we wished to analyze a data set that demonstrated an association between a phenotype and Shannon α diversity, which is related to the geometric mean as it is the logarithm of the inverse of the geometric mean of a sample that is weighted by the relative abundances (15). The biological relevance of α diversity to the microbiome has been extremely well documented across a wide range of scenarios (e.g., references 16–24). Specifically, we reanalyzed data from a study of the vaginal microbiome and preterm birth and examined the first time point from each of the 40 individuals included in the study (16) (see "Methods"). As previously reported (25–27), we observed an association between α diversity and subsequent preterm birth (Fig. 2a; Mann–Whitney $U$ $P = 0.037$).

We next selected a sparse taxon, which was only detected in a single sample: OTU 4465907, which was identified by the authors (16) as a *Blautia* sp. (Fig. 2b). When we performed a CLR transformation of this data set (see "Methods"), we noted that as expected, values were introduced to this feature, and because Shannon α diversity

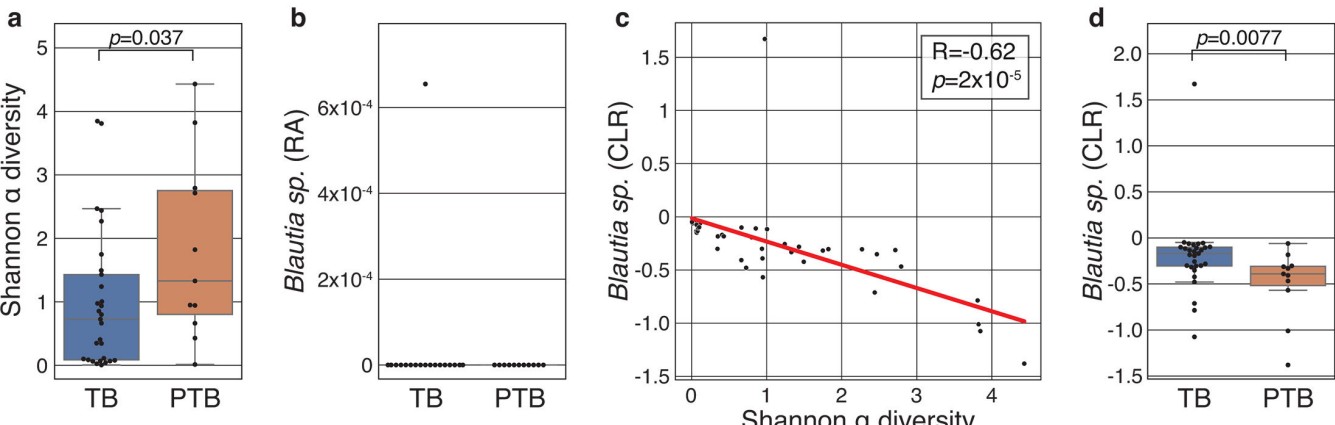

**FIG 2** CLR transformation generates associations between a sparse taxon and preterm birth. (a) Box and swarm plots of the α diversity of vaginal microbiome samples collected during pregnancy, separated by subsequent term (TB) or preterm birth (PTB). As was previously noted, the α diversity of the vaginal microbiome is associated with preterm birth (Mann–Whitney $U$ $P = 0.037$). (b) Box and swarm plots showing the relative abundance (RA) of a *Blautia* sp. (OTU 4465907), a sparse feature that was only detected in a single sample. (c) Scatterplot and fitted ordinary least squares curve of the same *Blautia* sp. that was CLR-transformed (*y*-axis) and the α diversity of the same sample (*x*-axis). Because the α diversity is related to the geometric mean (15), we observe a strong negative correlation (Pearson's $R = -0.62$, $P = 2.0 \times 10^{-5}$). (d) Same as panel b, showing the CLR-transformed relative abundances of the same *Blautia* sp. Since preterm birth is associated with α diversity and α diversity is negatively associated with the CLR-transformed sparse feature, the latter becomes associated with preterm birth (Mann–Whitney $U$ $P = 0.0077$). Box, IQR; line, median; whiskers, nearest point to 1.5*IQR.

is related to the geometric mean, the CLR-transformed *Blautia* sp. became strongly negatively correlated with the α diversity (Fig. 2c; Pearson's $R = -0.62$, $P = 2.0 \times 10^{-5}$). As a result, we now also observe a negative association between *Blautia* sp. and subsequent preterm birth (Fig. 2c; Mann–Whitney $U$ $P = 0.0077$). As before, this result holds across a wide range of pseudocounts ($P < 0.01$ for pseudocounts ranging from $10^{-8}$ to 0.01; Fig. S2b). Importantly, we note that once again, the compositional transformation of the sparse feature that we implemented is a simple sample-wise operation that is not observing any patient metadata or information from any other sample, and it is not erroneous to use this feature as a predictor for preterm birth. While it would be incorrect to interpret this association as indicating anything about the biology of *Blautia*, which is not necessarily even present in these vaginal samples, it would be accurate to interpret this as an indication of a "microbiome-wide" signature, in that the log of the inverse of geometric mean is associated with the outcome of interest. While these multivariate compositional transformations can introduce challenges with interpretation and inference, our results demonstrate one example of a valid explanation for why the transformation of an empty feature is associated with a biological phenotype on a real microbiome data set.

## The associations between tumor type and originally sparse genera highlighted by Gihawi et al. can be reasonably explained by a CLR transformation

Gihawi et al. (1) claimed that Poore et al. (3) had major errors in the data analysis of microbiome data from TCGA. Among other critiques, Gihawi et al. claimed that features that were very sparse (almost entirely zero) in the raw data were erroneously filled with an artificial tag that leaked prior information about the tumor type (1), which they supported with four specific examples. Above, we already showed, via counterexamples, that observing such associations is not sufficient to prove a data analysis error or leakage in machine learning analysis. Nevertheless, we wished to determine if Gihawi et al.'s observations could be explained by interpretable attributes of the samples' overall compositions, as opposed to an erroneous incorporation of sample metadata, as was

suggested. CLR is particularly relevant to the processing performed by Poore et al. because it bears similarity to voom (6). To this end, we reanalyzed the four examples Gihawi et al. provided in their Fig. 2–5 (see "Methods"). We note, however, that while this analysis evaluates this specific critique by Gihawi et al., it does not serve as a validation of the analysis by Poore et al. or of the normalization and batch-correction pipeline they used.

First, Gihawi et al. examined the values of *Hepandensovirus* in adrenocortical carcinoma (ACC). Upon reanalysis, we notice an underlying difference in α diversity between ACC and all other samples (Mann–Whitney $U$ $P = 2.6 \times 10^{-4}$; Fig. 3a). Despite the fact that *Hepandensovirus* is only detected in a single sample and is likely not truly present in tumors (2) (Fig. 3b), a CLR transformation of this feature results in values that are different in ACC compared to other samples (Fig. 3c; $P = 3.2 \times 10^{-14}$) because the α diversity is related to the geometric mean. Therefore, the same association highlighted by Gihawi et al. (reproduced in Fig. 3d) can be observed in a compositional transformation that we know cannot cause information leakage because it does not consider values from any other samples, nor any sample metadata, but only the values of the sample itself. Of course, this does not mean that *Hepandensovirus* should be interpreted as biologically relevant to the tumor microbiome. Instead, the multivariate transformation of the data generated an association that reflects a shift in the global microbiome composition of the sample.

Second, Gihawi et al. examined the values of *Thiorhodospira*, comparing whole-genome sequencing (WGS) data from primary kidney chromophobe (KICH) tumor samples and normal tissue samples from the same group of patients. Upon reanalysis, we find a similar trend: an underlying difference in the sample α diversity (Mann–Whitney $U$ $P = 0.013$; Fig. 3e) is related to the observation that despite being originally very sparse (Fig. 3f), this feature becomes highly associated with KICH following CLR transformation ($P = 3.4 \times 10^{-6}$; Fig. 3g). Similarly large differences were observed by Gihawi et al. (reproduced in Fig. 3h).

Third, Gihawi et al. examined the values of *Nitrospira*, comparing between lung squamous cell carcinoma (LUSC) samples and all other primary tumor samples. Once more, we observe a significant association of LUSC with sample α diversity (Mann–Whitney $U$ $P = 1.6 \times 10^{-31}$; Fig. 3i). Interestingly, while Gihawi et al. write that "in the raw data, there is no such shift" in *Nitrospira* (1), in this case, we saw a significant association with LUSC even when examining raw relative abundances ($P = 2.8 \times 10^{-141}$; Fig. 3j), a difference that was also present in the raw counts. While this difference in the relative abundances undermines our ability to attribute observed downstream differences solely to compositional transformations, we nevertheless continue to observe a significant association of the CLR-transformed values of *Nitrospira* with LUSC ($P = 1.1 \times 10^{-127}$; Fig. 3k). As before, we conclude, using the same features highlighted in their manuscript, that the observation by Gihawi et al. (reproduced in Fig. 3l) cannot be attributed to data analysis error or information leakage without additional support.

Fourth, Gihawi et al. examined the values of *Mulikevirus*, comparing data from primary head and neck squamous cell carcinoma (HNSC) tumor samples and normal tissue samples from the same group of patients. In this case, we did not find the same phenomenon of associations between sample α diversity and CLR-transformed *Mulikevirus* with HNSC (Fig. 3m through p). However, this still is not necessarily an indication of information leakage since we have only examined a single compositional transformation which is far less complex than the full set of transformations performed by Poore et al.

Finally, we note that while all our analyses so far used a single pseudocount that was introduced in relative abundance space (see "Methods"), with the goal of isolating the impact of CLR, we observe statistically significant differences in the same three out of four taxa when varying the choice of pseudocount across most values ranging from $10^{-8}$ to 0.01 (Fig. S2c through f). We observe even greater differences when introducing a pseudocount of 1 to the raw counts (Fig. S3) because this produces differences in the

## Tumor-associated values introduced into sparse features by Poore et al. can be reasonably explained by compositional transformations

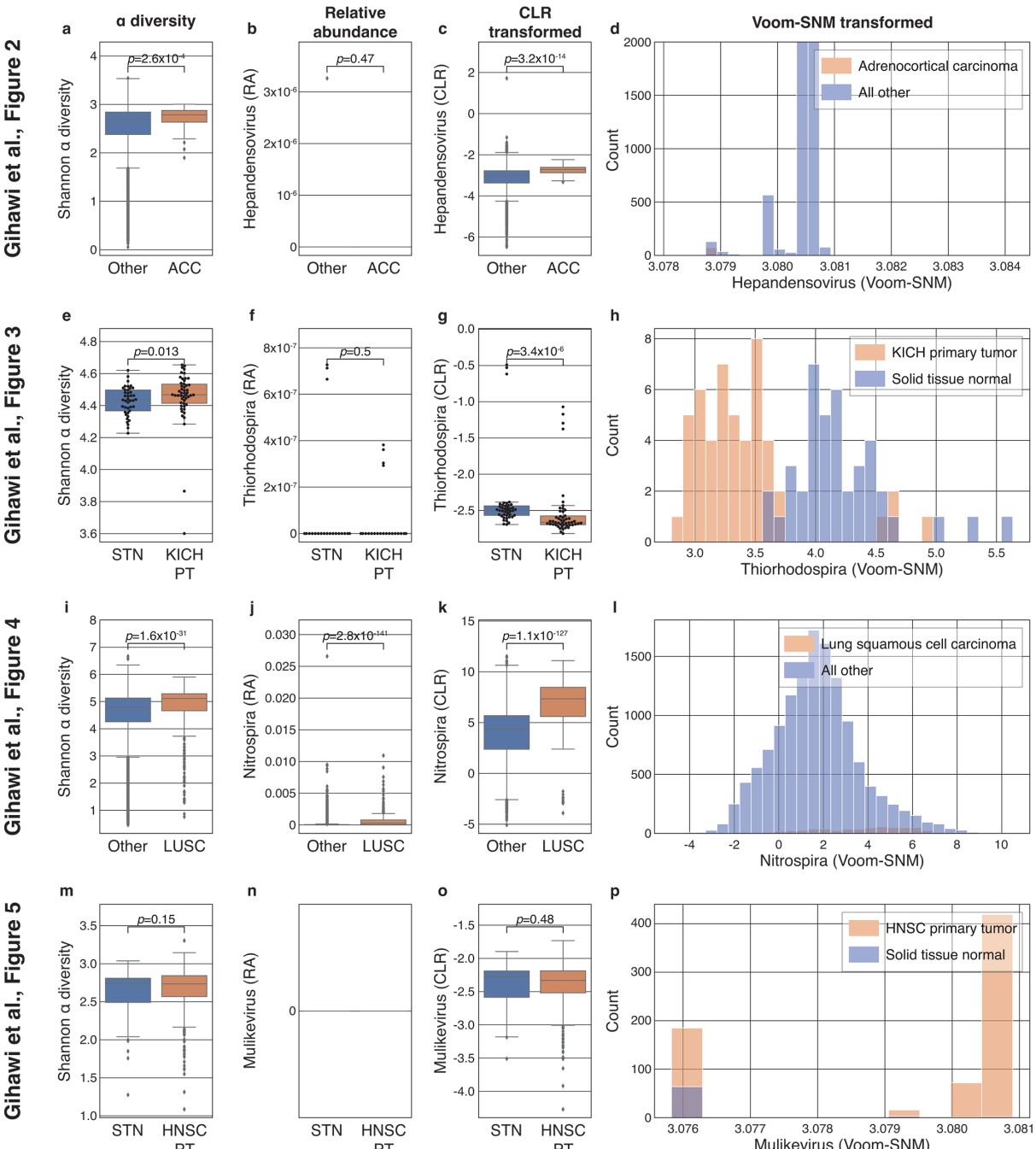

**FIG 3** CLR transformation can reproduce predictive sparse features highlighted by Gihawi et al. Each row shows the same analyses applied to a different scenario analyzed by Gihawi et al. (1) in their Fig. 2–5, respectively (see "Methods"). (a, e, i, m) Boxplots of the α diversity of samples. (b, f, j, n) Boxplots of the relative abundance (RA) of the taxon analyzed. (c, g, k, o) The CLR-transformed values of the same taxa. (d, h, l, p) Histograms replicating Fig. 2–5, respectively, in Gihawi et al. (1) with some differences noted in "Methods." In the three cases in which significant differences were observed in α diversity (a, e, i), the CLR transformation produced tumor type-associated values (o, g, k) from originally sparse features (b, f, j). Box, IQR; line, median; whiskers, nearest point to 1.5*IQR. Individual dots are plotted if ≤100 samples per plot, only outliers are displayed if >100 samples.

relative abundances of sparse features as a result of different underlying read counts—
also commonly associated with sample-wise factors such as α diversity.

## Gihawi et al. did not perform an information-free analysis

Gihawi et al. conclude their analysis of sparse features by performing a classification analysis on "information-free raw data" (1). For completeness, we note that while it may appear that this analysis includes the application of a machine learning pipeline to a matrix of zeros, this was not the analysis performed. As Gihawi et al. specify: "We then populated each cell in the empty matrix with its corresponding value from the Voom-SNM normalized data" (1), indicating that the analysis performed was of the voom-SNM normalized data that was subset to features that were originally zero. We note that this corresponds to an analysis of the empty feature in our simulated data set (Fig. 1e), which, as we showed, contains legitimate information that can perfectly classify the label. These conclusions made by Gihawi et al. rely upon the assumption that all zeros in microbiome data should be interpreted identically. However, this assumption does not always hold for microbiome data sets, for which certain features can be zero due to undersampling, while some are structural zeros. Thus, it is common for transformations of microbiome data sets to encode additional information into features that were initially sparse, even if it is simply based on numerical summaries of the remainder of the corresponding sample, as in the CLR transform. Therefore, it is important to interpret microbiome features as multivariate. Taken together with other results presented here, we thus show that this analysis by Gihawi et al. is insufficient to demonstrate flaws in the normalization process used by Poore et al.

## Discussion

Determining the validity of a transformation that induces associations with a phenotype in non-informative features bears importance for the evaluation of machine learning pipelines. We present three analyses demonstrating that such a phenomenon is an expected outcome of a commonly used transformation, the CLR transformation, especially for cases in which the phenotype is associated with the geometric mean of the taxonomic composition or its α diversity. As the CLR is a sample-wise operation, which does not use an outcome label or information from other samples, it cannot independently introduce information leakage. Through these counter-examples, we therefore demonstrate that observing phenotype-associated values in previously sparse features should not be considered sufficient to conclude that an analysis suffered from an artificial inflation of predictive signals.

Our reanalysis of the information leakage claims raised by Gihawi et al. strongly suggests that their observations are an expected and reasonable effect of using voom-SNM, a compositional transformation that performs linear adjustments in log-space, especially as we observed an association between the relevant phenotype and α diversity in most cases. Nevertheless, this work does not serve to validate the tumor microbiome associations reported by Poore et al. (3, 28), nor their normalization or batch-correction pipeline. We therefore consider other concerns raised by Gihawi et al., such as potential contamination (2) or errors in the genome database used leading to misclassification of human reads as bacteria (1), out of scope for this analysis. Our conclusion is that Gihawi et al. did not provide robust evidence of an artificial tag, information leakage, or any synthetic inflation of predictive results in the analysis conducted by Poore et al. (3). However, we did not demonstrate that such leakage did not exist. Furthermore, both our and Gihawi et al.'s analyses demonstrate challenges in the biological interpretation of univariate associations detected with specific taxa.

The pivotal assumption underlying Gihawi et al.'s analysis and conclusion is that features in microbiome samples are univariate and independent of each other. However, because microbiome samples are compositional, the abundance of any taxon can only be evaluated within the context of the measured abundances of other taxa within the same sample. It therefore should be considered reasonable to encode sample-wide information into transformations of features observed within multivariate and compositional data, such as microbiome data. This implies that values measured as zeros in microbiome data do not necessarily mean the same for different samples, for example,

due to different sequencing depths and α diversity. Additionally, some zeros may be structural (microbe is not present in the ecosystem) as opposed to sampling zeros (microbe is present in very low abundance). Such distinctions are an ongoing challenge in the field, which are important for interpretation, but are independent from the issue of information leakage in this case. Therefore, evaluating if features which are initially zero are transformed into distinct values should generally be interpreted from a multivariate perspective in microbiome data, and observing phenotype-associated values should not be considered proof of a data analysis error.

There are more direct ways to check whether a machine learning pipeline is invalid or "leaks" information than examining specific features. Our recommendation, for any machine learning analysis, is to perform a permutation test in which the phenotype labels are shuffled before running any data normalization and machine learning pipelines. Under such analyses, a machine learning model is expected to yield random predictions, and any observed signal is a strong indication of information leakage. We and others have also previously noted that predictive analyses are more robust if cross-validation is done across batches (4, 29).

Finally, we note that there exist many reasonable variations to the data processing steps that we used in this work. We have chosen to use the CLR transformation in this work due to its simplicity, widespread use, and the fact that it is strictly sample-wise. We note, however, that it does have its downsides as it imposes constraints on the data that make strong implicit assumptions and are still subject to a sum constraint. Alternatives to compositional transformations that instead model scale uncertainty have recently been proposed (12, 30). We also note that while we evaluated a few variations of pseudocount methods across which our results are robust (Fig. S2 and S3), many alternative strategies are available, including some that do not require pseudocounts (31, 32). However, this does not change our conclusions as the processing choices used in this work do not induce any sort of information leakage and are therefore sufficient to serve as a counter-example to the claims by Gihawi et al. (1). Since there exists at least one reasonable transformation that introduces phenotype-associated values into sparse features, observing such a case should not be considered independently sufficient to challenge the results of a predictive pipeline.

## Methods

### Synthetic simulations

We generated a simple data set of 100 synthetic samples, of which 50 had a positive phenotype label and 50 negative (Fig. 1b). Among the 50 positive samples, we simulated data by drawing 30 features from i.i.d. uniform distributions in [0,1], in addition to a 31st empty feature of zero counts for all samples. To simulate different geometric means for the samples with negative phenotypes, we drew 20 features from i.i.d. uniform distributions in [0,1], with the remaining 11 features being empty. By construction, this leaves the 31st feature as "0" for all samples (Fig. 1a). We then transformed to relative abundance space, such that all rows summed to 1, added a pseudocount of $10^{-6}$ to all samples, corresponding to 10 to the largest power that keeps the pseudocount below the data set's smallest observed relative abundance value. We then applied the CLR transformation using the "skbio.stats.composition.clr" function (33) (Fig. 1d). Following this transformation, we examined the association of the originally zero 31st feature with the simulated labels (Fig. 1e). We note that our observed differences would be larger if we added the pseudocount before transforming to relative abundance space, although doing so would induce differences already in the relative abundance table prior to the CLR transformation, which would confound our conclusions. For all analyses, we re-ran the same pipelines using pseudocounts ranging across all powers of 10 from $10^{-8}$ to 0.01 (Fig. S2).

## Analysis of the vaginal microbiome and preterm birth

We obtained publicly available microbiome read counts and metadata from Data sets S1 and S2 of a study profiling the vaginal microbiome along 40 pregnancies (16). We filtered this data set to consider only the first vaginal sample collected from each patient, and among those, kept the 222 taxa with at least one read observed in a sample. We considered a sparse feature, OTU 4465907, which was identified by the authors as a *Blautia* sp. and which was only observed in a single sample (Fig. 2b). We transformed the data to relative abundance space; added a pseudocount to all samples using the same strategy as before, which yielded a pseudocount of $10^{-4}$; and then ran the "skbio.stats.composition.clr" function. We then compared the associations of α diversities, clr-transformed relative abundances of *Blautia* sp., and preterm birth using, as applicable, Mann–Whitney $U$ and Pearson's $R$.

## Reanalysis of The Cancer Genome Atlas reanalysis

We obtained publicly available microbiome read counts and metadata from the original analysis by Poore et al. (3) in order to reanalyze the four examples highlighted in the second critique by Gihawi et al. (1). We made our best effort to match the analyses performed by Gihawi et al. (1) by reproducing their histograms (Fig. 3d, h, l, and p). We performed the following analyses:

i. *Hepandensovirus* and ACC (Fig. 2 in Gihawi et al. [1]; our Fig. 3a through d; Fig. S2c and S3a and b). We analyzed the "Most Stringent Decontamination" file of Poore et al. (3), comparing ACC samples to all other tumor types. To match the appearance of Fig. 3d to Fig. 2 in the study by Gihawi et al., we artificially limited the upper limit of the *x*-axis to 3.0845, which removed one outlying point, and the upper limit of the *y*-axis to 2,000.

ii. *Thiorhodospira* and KICH (Fig. 3 in Gihawi et al. [1]; our Fig. 3e through h; Fig. S2d and S3c and d). We analyzed the "All Putative Contaminants Removed" file of Poore et al. (3), comparing KICH primary tumor WGS data to normal tissue samples from patients with KICH.

iii. *Nitrospira* and LUSC (Fig. 4 in Gihawi et al. [1]; our Fig. 3i through l; Fig. S2 and S3e and f ). We analyzed the "All Putative Contaminants Removed" file of Poore et al. (3), comparing LUSC samples to all other tumor types.

iv. *Mulikevirus* and HNSC (Fig. 5 in Gihawi et al. [1] our Fig. 3m through p; Fig. S2 and 3g and h ). We analyzed the "Most Stringent Decontamination" file of Poore et al. (3), comparing data from HNSC primary tumor to normal tissue samples from patients with HNSC. We note that Fig. 3p is different from Fig. 5 in Gihawi et al. (1) because we maintained default plotting parameters to display visually consistent spacings throughout the entire plot.

For each of the four analyses, we obtained the raw read counts for the corresponding samples, removed features marked as "contaminants," and measured their Shannon diversities (Fig. 3a, e, i, and m), relative abundances (Fig. 3b, f, j, and n), and CLR of the relative abundances following either a pseudocount of 10 to the largest power that keeps the pseudocount below the data set's smallest observed relative abundance value (Fig. 3c, g, k, and o) or a pseudocount of 1 introduced before relative abundance normalization (Fig. S3). Mann–Whitney $U$ test was used for all pairwise comparisons.

## ACKNOWLEDGMENTS

We thank members of the Korem group for useful discussions. We thank all authors and participants involved in the generation of data used in this study.

This work was supported by the Program for Mathematical Genomics at Columbia University (T.K.) and T15LM007079 (G.I.A.).

G.I.A. and T.K. conceived and designed the study, designed analyses, interpreted the results, and wrote the manuscript. G.I.A. conducted all analyses.

## AUTHOR AFFILIATIONS

[1]Department of Biomedical Informatics, Columbia University Irving Medical, New York, New York, USA

[2]Program for Mathematical Genomics, Department of Systems Biology, Columbia University Irving Medical Center, New York, New York, USA

[3]Department of Obstetrics and Gynecology, Columbia University Irving Medical Center, New York, New York, USA

## AUTHOR ORCIDs

George I. Austin http://orcid.org/0000-0002-7834-4968
Tal Korem http://orcid.org/0000-0002-0609-0858

## FUNDING

| Funder | Grant(s) | Author(s) |
| --- | --- | --- |
| U.S. National Library of Medicine | T15LM007079 | George I. Austin |

## DATA AVAILABILITY

The vaginal microbiome data used in this analysis are available in Datasets S1 and S2 of reference 16. TCGA data used in this analysis are available from reference 2. All code used in this analysis is available at https://github.com/korem-lab/compositional-empty-features.

## ADDITIONAL FILES

The following material is available online.

### Supplemental Material

**Supplemental Material (mSystems00021-25-s0001.pdf).** Fig. S1-S3.

### Open Peer Review

**PEER REVIEW HISTORY (review-history.pdf).** An accounting of the reviewer comments and feedback.

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
