## [Reviewer comments · mSystems]

Compositional transformations can reasonably introduce phenotype-associated values into sparse features

George Austin and Tal Korem

Corresponding Author(s): Tal Korem, Columbia University

Review Timeline:

Submission Date:

January 13, 2025

Accepted:

February 13, 2025

Editor: Christian Diener

Reviewer(s): Disclosure of reviewer identity is with reference to reviewer comments included in decision letter(s). The following individuals involved in review of your submission have agreed to reveal their identity: Justin D Silverman (Reviewer #2)

Transaction Report:

DOI: <https://doi.org/10.1128/msystems.00021-25>

Re: mSystems00021-25 (Compositional transformations can reasonably introduce phenotype-associated values into sparse features)

Dear Dr. Tal Korem:

Thank you for your patience and addressing all the reviewer comments. Both reviewers are content with the revisions and had no further comments. Congratulations on your manuscript that has now been converted to a Matters Arising as per your request.

Your manuscript has been accepted, and I am forwarding it to the ASM production staff for publication. Your paper will first be checked to make sure all elements meet the technical requirements. ASM staff will contact you if anything needs to be revised before copyediting and production can begin. Otherwise, you will be notified when your proofs are ready to be viewed.

Sincerely,
Christian Diener
Editor
mSystems

Reviewer #1 (Comments for the Author):

The authors have addressed all my concerns. I appreciate their detailed responses to my comments and revising the manuscript accordingly. I have no further comments.

Reviewer #2 (Comments for the Author):

The authors have addressed my concerns. I thank them for their contribution to the field.